# A Portable Continuous-Flow Polymerase Chain Reaction Chip Device Integrated with Arduino Boards for Detecting *Colla corii asini*

**DOI:** 10.3390/mi13081289

**Published:** 2022-08-11

**Authors:** Shyang-Chwen Sheu, Yi-Syuan Song, Jyh-Jian Chen

**Affiliations:** 1Department of Food Science, National Pingtung University of Science and Technology, 1, Shuefu Road, Neipu, Pingtung 91201, Taiwan; 2Department of Biomechatronics Engineering, National Pingtung University of Science and Technology, 1, Shuefu Road, Neipu, Pingtung 91201, Taiwan

**Keywords:** *Colla corii asini*, polymerase chain reaction, food security, Arduino, portable device

## Abstract

Food security is a significant issue in modern society. Because morphological characters are not reliable enough to distinguish authentic traditional Chinese medicines, it is essential to establish an effective and applicable method to identify them to protect people’s health. Due to the expensive cost of the manufacturing process and the large volume of the analytical system, the need to build a portable and cheap device is urgent. This work describes the development of a portable nucleic acid amplification device integrated with thermal control and liquid pumping connecting to Arduino boards. We present a novel microfluidic polymerase chain reaction (PCR) chip with symmetric isothermal zones. The total chip volume is small, and only one Arduino board is needed for thermal control. We assemble a miniaturized liquid pump and program an Arduino file to push the sample mixture into the chip to implement the PCR process. In the proposed operation, the Nusselt number of the sample flow is less than one, and the heat transfer is conduction only. Then we can ensure temperature uniformity in specific reaction regions. A *Colla corii asini* DNA segment of 200 bp is amplified to evaluate the PCR performance under the various operational parameters. The initial concentration for accomplishing the PCR process is at least 20 ng/μL at the flow rate of 0.4 μL/min in the portable continuous flow PCR (CFPCR) device. To our knowledge, our group is the first to introduce Arduino boards into the heat control and sample pumping modules for a CFPCR device.

## 1. Introduction

Food security is one of the most critical issues in modern society. People eat food not just for their appetite as humans but also for their physical and mental health. Some valuable traditional Chinese medicines (TCMs) have been widely used as tonic food for a long time, especially in some East Asian countries. However, as the demand for TCMs has risen rapidly over the past decades, some substitutions derived from the skins or bones of animals have been fraudulently labeled as true TCMs. These substitutions sometimes have little pharmaceutical effect and harm people’s health seriously. Because morphological characters are not reliable enough to distinguish true TCMs, it is essential to establish an effective and applicable method to identify them to protect people’s health. People have the right to choose healthy food. Therefore, the identification of the food for safety purposes is very significant [1].

It is critical to establish a reliable and convenient method for food identification. The bacterial culture method is considered the gold standard for diagnosis in clinical practice for food security. It is specific and accurate. However, it is also time-consuming, complex, and without high sensitivity [2]. Some researchers have also demonstrated various spectroscopy-based and chromatographic methods for quality control [3,4]. However, similar chemical properties in an adulterant sometimes make it difficult to identify species, and they have not reached the stage of practical application due to sample-processing variability. An alternative method for specific analysis in species identification is the nucleic acid-based assay [5], which became popular during the last decades.

Deoxyribonucleic acid (DNA)-based molecular diagnostics have been widely applied in order to authenticate species for health and religious purposes in food security. Since Saiki et al. [6] invented polymerase chain reaction (PCR) in the 1980s, PCR has become one of the most established molecular biology techniques in recent decades. Lots of researchers utilized PCR in the biotechnology field, such as in criminal forensics, genetic analysis, and medical diagnostics, due to the exponential amplification of tiny amounts of specific DNA molecules to a detectable concentration. The PCR process goes through the following stages. Firstly, double-stranded DNA (dsDNA) is separated into two single-stranded DNA (ssDNA) at high temperatures (denaturation stage at about 95 °C or 368 K). Secondly, primers bind to their complementary site of ssDNA at low temperatures (annealing stage at about 55 °C or 328 K). Thirdly, the thermostable DNA polymerase extends the primers, complementary to the DNA template, at intermediate temperatures (extension stage at about 72 °C or 345 K).

Conventional PCR machines are sizable and require a long cycling time. During the last three decades, microelectromechanical systems (MEMS) fabrication processes play an essential role in reducing these inconveniences. Implementation of such miniaturized devices enables the development of fast, easy-to-use, and portable systems with a high level of automation and functional integration for applications such as point-of-care (POC) diagnostics. Moreover, the disposable elements used in the miniaturized devices prevent the samples from cross-contamination.

Many researchers divided the micro-PCR (μPCR) devices into two types based on the handling of sample mixtures. One is the static chamber μPCR devices. The injected sample and the whole reaction chamber are stationary in the static chamber device and undergo external thermal variations to accomplish the PCR process. These devices are a scaled-down version of commercial thermocyclers and perform repeated heating and cooling. Any enzyme adsorbed by the walls of the reaction volume is slight. Kaprou et al. [7] designed the embedded resistive microheaters for realization in the inner Cu layer of the commercially available PCB substrate. Huang et al. [8] developed a microfluidic chamber-based PCR-array system. The system can screen multiple respiratory pathogens in an integrated manner. There is a Peltier heating device responsible for the thermal control of PCR.

On the contrary, the PCR mixture flows consecutively across the distinct heated zones corresponding to the temperature steps needed for PCR in continuous-flow (CF) PCR devices. The reaction mixture moves through the individual isothermal zones by keeping the temperatures constant over time at different locations in the device. The arrangement of isothermal zones determines the residence time and the number of thermal cycles [9].

The usual method of arranging the heater location in CFPCR devices is to place some heaters side by side. Then, the sample mixture flows through the required temperature regions along a serpentine channel to accomplish the PCR process. Ragsdale et al. [10] presented a disposable polycarbonate (PC) device that included the amplification of the foot-and-mouth disease virus (FMDV)-derived cDNA. The opposite sides of the chip demonstrate the placement of two thin-film heaters. The PCR mix was pulled through the chip using a commercial syringe pump. Li et al. [11] explored a glass-polydimethylsiloxane (PDMS) bonding chip for multi-PCR of *Porphyromonas gingivalis*, *Treponema denticola*, and *Tannerella forsythia*. The chip system is composed of a commercial syringe pump and two positive temperature coefficient (PTC) ceramic heaters controlled by two commercial temperature controllers. Jiang et al. [12] developed a microfluidic device combining CFPCR and DNA hybridization to detect bacterial pathogens. Using a polyimide heating membrane, they built a thermal cycler of the PDMS-based device. A syringe pump connected to the inlet and outlet of the microchannel implemented the sample injection and collection, respectively. Chang and You [13] presented an irreversible bond between a PDMS microfluidic chip and a half-cured PDMS film coated on a PCB substrate. The PCB substrate with three heating electrodes supplied the specific heating zones and amplified a 120-bp DNA fragment from BEAS-2B of human bronchial epithelial cells. Pješčić et al. [14] utilized a commercial syringe pump to inject the phage DNA template (ΦX174) sample into the microfluidic channel. The authors designed the thermal controlling system comprised of a cartridge heater, a heat sink, and a PID controller to generate high- and low-temperature regions within the glass-composite device. Schaerli et al. [15] presented an SU-8 substrate device for sample droplets. A cartridge heater and a Peltier module supported the denaturation and annealing zones of the device, respectively. An 85 bp sequence of the tyrocidine synthetase 1 gene (P09095) solution was in syringes pumped using the Harvard Apparatus pump. Fukuba et al. [16] demonstrated a microfabricated device in environmental microbiology. Six indium tin oxide (ITO) heaters on a glass substrate define three isothermal zones for amplifying 580 and 1450 bp of DNA fragments. Hsieh et al. [17] presented the integrated PMMA chip combined with a microfluidic PCR and a surface plasmon resonance sensor to detect *LMP1* DNA. Two aluminum blocks with electric heating films and PID controllers defined the temperature zones.

Some researchers choose the other heater arrangement and distribute the heating regions circumferentially to avoid unnecessary heating during PCR. The reactant plug moves through a spiral microchannel or tube to the outlet to accomplish the PCR process. Peham et al. [18] demonstrated a cyclic microsystem for amplifying the 16S ribosomal RNA gene. The miniaturized device consists of three Peltier elements driven by commercial PID controllers and arranged to form a triangular prism. The microfluidic tubing comprises transparent polytetrafluoroethylene (PTFE) and is coiled around the triangular prism 40 times. Chung et al. [19] presented a polymer disk equipped with a spiral microchannel. Six temperature controllers maintained the temperatures of the metal plates at different constant values. The syringe pump pushed a transport liquid containing the fragment from arylamine N-acetyltransferase DNA through a PTFE capillary tube. Shu et al. [20] proposed a segmented CD-PCR on a spiral PTFE tubing microfluidic device. The device for identifying four foodborne bacterial pathogens mainly consists of three resistance cartridge heaters, a LabVIEW-based temperature control system, and a syringe pump. Kim et al. [21] developed a cylindrical device combined with a laser-induced fluorescence detection system for real-time PCR. They chose a high-purity perfluoroalkoxy alkane (PFA) capillary as the reactor. The device includes three holes for the cartridge heaters controlled by a series of commercial controllers. Hajji et al. [22] presented a microfluidic PCR platform in which sample droplets are transported in a PTFE capillary by the syringe pump and flow through two isothermal areas. Peltier modules were used for temperature control by independent proportional–integral–derivative (PID) boards.

Typically, microfluidic systems transport fluids and the fluid pump operates in most applications. However, pumping devices may lead to a costly system. Researchers seek to make a simple and easy pump and integrate it into the CFPCR system. Wu et al. [23] proposed a self-activated micropump system for applications of on-chip PCRs. A single heater achieves the thermal cycle requirement by controlling the temperature gradient of the PDMS block. Nagai and Fuchiwaki [24] described a PCR system to detect a microorganism modeling anthrax and a point mutation of the *FGFR3* gene. They integrated two cartridge heaters, a self-made syringe pump, a fluorescence detector, and a tablet personal computer into a suitcase.

Commercial PID controllers provide a total solution for temperature control. In portable devices, it is convenient to utilize a microcontroller which is often low-powered consumption. Han et al. [25] fabricated a PDMS-based chip and indium tin oxide (ITO) coated heaters on a CFPCR platform. The heating circuit used an ATMega 128 MCU (Microchip Technology, Chandler, AZ, USA) to control the temperature. A syringe pump injects the PCR mixture into the microchannel. Kulkarni and Goel [26] developed a PDMS–glass bonding chip with a serpentine microchannel. The NodeMCU microcontroller (an open source firmware) was responsible for the temperature control of the cartridge heater. A commercial syringe pump pushed the reaction volume through the channel. Finally, the device amplified the rat *GAPDH* gene of the 594-bp successfully. Talebi et al. [27] presented a CF microfluidic device for PCR and fabricated the heater electrodes based on PCB technology. They used the Arduino MEGA to control the heater temperatures. The PCR mixture was pumped into the PMMA microchannel using an ordinary syringe pump.

During the last few decades, microfluidic PCR chips have gained much attention due to their small thermal masses and the high heat transfer rate inside the system. A CFPCR device can regulate the temperature of the reactants by moving the reactants through different temperature regions. The merit of establishing several isothermal areas for CFPCR simplifies the fabricated cost and reduces the research and design cycle time. Some researchers have designed a five-temperature-region chip with a serpentine channel for CFPCR [27,28]. It does reduce the total volume of the chip device. External pumping in the microfluidics field is an essential function for sample transportation. However, a standard syringe pump is expensive and occupies the majority of the system cost. Building a user-friendly and miniaturized syringe pump for the microfluidic device is one of the cost-saving measures in the portable CFPCR device.

Due to the high demand for valuable gelatinous TCMs, bovine or swine skin was often used to make fake or adulterated *Colla corii asini* in recent decades. This situation exposes public health to high risk and causes unfair competition in the commercial markets. Establishing a reliable and convenient technology to develop an accurate detection method and identify fake health food before being used is essential. This work describes a portable nucleic acid amplification device integrated with thermal control and liquid pumping connecting to Arduino boards. We present a novel microfluidic PCR chip with symmetric isothermal zones. The total chip volume is small, and only one Arduino board is needed for thermal control. We assemble a miniaturized liquid pump and program an Arduino file to push the sample mixture into the chip to implement the PCR process. The use of Arduino has increased exponentially during the last two decades due to its readability and easiness [29,30]. The main contribution of the current work is to integrate the thermal control and liquid pumping modules into the microfluidics system and test the PCR performance of the CFPCR device by incorporating two Arduino boards. We can greatly decrease the device volume, save the system cost, and reduce the research and design cycle time. One heater supports the denaturation zone located at the center of the chip. A Peltier element cools the annealing zones set at the two sides of the chip. A DC power supported by three serial-connected 18,650 batteries or another 12 V external DC power supplies the power for performing CFPCR. A *Colla corii asini* DNA segment of 200 bp is amplified to evaluate the PCR performance under the various operational parameters. The specific amplification products in our device and the thermal cycler are clear. In the future, we can apply the portable continuous-flow device to a low-cost PCR system.

In the following, we first describe our main methods regarding the design of the microfluidic chips, the fabrication of the heating blocks and thermal control modules, the assembly of the homemade syringe pump, the development of the power supply and user interface, the surface treatment of the microchannel, and the operation of the device. Then, we express the characteristic results of the homemade pump. Next, we investigate the influences of the operational parameters of the system on the temperature uniformity of the chip. Finally, we conduct some experimental work to complete the contents of this article.

## 2. Materials and Experimentation

The portable microfluidic PCR device comprises a microfluidic chip, a set of thermal control module, a home-made syringe pump, a power supply module, and a user interface module. The DNA mixture pumped through a serpentine channel with various widths meanders inside the chip. Only one heater supports the denaturation zone located at the center of the chip. A Peltier element cools the annealing zones set at the two sides of the chip. The temperature distribution is created within the chip so that the DNA mixture can be heated and cooled during several thermal cycles to finish the PCR process. A syringe pump continuously pushes a sample mixture of a specific volume in a glass syringe at a fixed flow rate through a Teflon capillary tube connected to the inlet of the chip. A schematic diagram and photographic image of the portable CFPCR device are shown in Figure 1.

### 2.1. Design of the Microfluidic Chips

The CFPCR chip that holds the sample during the PCR process comprises a PDMS layer and a glass slide with thicknesses of 1.9 and 1.1 mm, respectively, as shown in Figure 2a. The outer dimensions of the PDMS layer are 40 mm in length and 25 mm in width. Those of the glass slide are 75 mm by 25 mm, similar to the size of a microscopy slide. The 30-loop channel of 50 μm depth is 200 μm in width, except for the extension region with a maximum width of 500 μm, as shown in Figure 2b. The pre-denaturation part is located near the inlet at the left-hand side of the chip. The pro-extension part is near the outlet at the right, shown in Figure 2c. After completing the SU-8 patterned master, we can use it to replicate the PDMS microchannel. It can considerably reduce the fabrication cost. Due to the low cost of the fabricated chip and the prevention of the sample cross-contamination between reactions, the reactor chip is disposable after use.

In our earlier work, we have expressed the comprehensive fabrication process of the PDMS-glass bonding chip [31]. The schematic diagram of the fabrication process is illustrated in Figure 3. Initially, a silicon wafer is cleaned and dehydrated on a hotplate. An epoxy-based negative photoresist (SU-8 3035) is spin-coated on a silicon wafer. We utilize UV exposure for the production of the channel pattern. After development, the master is washed and baked to fix the photoresist. The PDMS (Sylgard 184, Dow Silicones Corp., Midland, MI, USA) mixture thoroughly mixed with the precursor and curing agent in a 10:1 weight ratio is degassed with a mechanical vacuum pump to remove air bubbles. After pouring the PDMS mixture onto the SU-8 patterned master, we use a convection oven to cure the PDMS. The replicas are peeled off carefully from the master.

The single-hole puncher punches the inlet and outlet holes on the PDMS channel chip. After surface oxidation and bonding, the designed microfluidic chips are ready. The pattern defines the number of cycles performed through the chip. The arrangement of the channel layout and the flow rate of the mixture determine the reaction time for each PCR step.

### 2.2. Fabrication of the Heating Blocks and Thermal Control Modules

Figure 4a shows the structure of the thermal cycling apparatus consisting of a cartridge heater, a Peltier element, aluminum fins, and thermally conductive aluminum blocks. When performing PCR, the PCR chip is attached tightly to the top side of the aluminum blocks. The aluminum heating blocks and the PDMS-based chip are assembled and fixed onto a polymethyl methacrylate (PMMA) housing as shown in Figure 4b. A machined PMMA block holds the microfluidic chip, rectangular and U-squared aluminum heating blocks. The heating blocks are isolated by a 2 mm air gap and set at different temperatures for denaturation and annealing.

In our designed chip, the denaturation region heated by one heater locates at the center of the chip, and the annealing regions supported by a Peltier element lie at the two sides of the chip. The extension regions span from the annealing region to the denaturation region. A PCR thermal cycle is successful when the reagent leaves the extension region. The symmetric management creates the five reaction regions in the chip. The five-temperature-region design requires only one half-loop per PCR cycle, as shown in Figure 4c. The fabrication of the half-size chip volume is mandatory. The thermal control modules are detachable from the channel chip and reused.

The photograph of the thermal control module is shown in Figure 5a and the functions are illustrated in Figure 5b. The cartridge heater (3.175 mm diameter, 38 mm length, 10 V, 14 W, C1J-9412, Watlow, St. Louis, MO, USA) inserted into the central part of the rectangular block (6 mm × 6 mm × 40 mm) keeps the chip temperature at the denaturation region. The temperature distribution inside the heating block is made highly uniform by the high thermal conductivity of aluminum. The temperature sensor, DS18B20 (Dallas Semiconductor, Dallas, TX, USA), mounted onto the aluminum heating block, is utilized to supply temperature feedback under a homemade PID controller. The contacted Peltier element module and the other sensor with a similar thermal control program regulate the annealing temperature. An aluminum block (40 mm × 5 mm × 27 mm) of a U-squared shape sticks on the surface of a Peltier element (4 mm × 4 mm × 3.5 mm, TEC1-12706, Hebei IT, Shanghai, China). An aluminum fin is placed under the Peltier element to achieve the required temperature at the annealing regions. The block is in contact with the chip via some thermal conductive adhesive. The temperature difference of the block surface at three measured points is about ±1 °C.

A schematic block diagram of the temperature controller is presented in Figure 5c. The Arduino Mega 2560 (an open source firmware) is a microcontroller board based on the ATmega2560 (Microchip Technology Inc., Chandler, AZ, USA) and is utilized for controlling the temperature of the heating blocks. An NPN (negative-positive-negative) Darlington power transistor (TIP120, STMicroelectronics, Geneva, Switzerland) regulates the heater power to heat the aluminum blocks. A single intelligent temperature sensor (DS18B20, Dallas semiconductor, Dallas, TX, USA) provides the temperature value for feedback control. Then, a 16 × 2 I2C LCD, which consists of an HD44780 (Hitachi, Tokyo, Japan)-based character LCD and an I2C LCD adapter, provides the user interface. A DC power supported by three serial-connected 18650 batteries supplies the power requirement when the system is working. The power supply part uses an LM2596S (Texas Instruments, Dallas, TX, USA) DC-DC buck converter step-down module to provide up to 3 A output current. The push buttons start the command given after entering the flow rate and timer values; the start function begins running the machine and the stop function is to stop all programs.

A program in the C++ programming language is compiled in an Arduino integrated development environment (IDE) for developing ATmega2560 microcontroller offerings and applications. The thermal sensors connect to Arduino Mega 2560. The microcontroller receives the temperature signals. The measured temperature compares with the setting temperature input using the buttons. The control signal determines the power inputs to the heater and the Peltier element using a proportional–integral–derivative (PID) algorithm. After the injection of the sample, the heater and the Peltier element maintain at specific temperatures.

### 2.3. Assembly of the Homemade Syringe Pump

A syringe pump injects the inserted fluid at a constant rate for some time. It also combines mechanical and electronic components that manipulate a standard syringe. Figure 6a demonstrates the homemade pumping unit. It is mainly composed of a stepper motor (14PM-M011-G1ST, Minebea Co., Ltd., Nagano, Japan), a lead screw (5 mm diameter, 100 mm length), a coupling (3 mm to 5 mm), two linear bearings (6 mm I.D., 12 mm O.D., 19 mm width), and two guide rods (6 mm diameter, 110 mm length). Some 5 mm thickness PMMA plates assemble the pump frame. It can substitute for the costly external precision syringe pump. In our work, a homemade syringe pump costs less than 200 USD and pumps the DNA mixture into the chip.

Figure 6b illustrates the block diagram of the syringe pump in our research. The construction is a DC voltage source for the power required by the Arduino Nano (an open source firmware) and the stepper motor, the buttons as an input interface of parameters, the Arduino Nano as a controller of all systems, ULN2003A (STMicroelectronics, Geneva, Switzerland) as the driver of the stepper motor, stepper motor as the lead screw pusher, LCD as a display for volumetric flow rate, and PMMA plates as a box from the syringe pump.

Figure 6c shows the circuit diagram of the syringe pump based on Arduino. Three serial-connected 18650 batteries supply the power requirement for the Arduino Nano and the stepper motor when the system is working. The LM2596S can take an input voltage of 12.6 V and convert it to 5 V up to 3A of continuous current. The Arduino Nano regulates the entire work process of the system by processing the flow rate entered through a push-button. The ULN2003A is the motor driver for 14PM-M011-G1ST stepper motor (MinebeaMitsumi, Inc., Nagano, Japan). The LCD shows information about the amount of flow rate.

The serial peripheral interface (SPI) is a synchronous serial communication interface specification used for short-distance communication, primarily in embedded systems. In this work, we use SPI protocol for communication between Arduino Mega and Arduino Nano. The Arduino Mega acts as Master, and the Arduino Nano acts as Slave.

A homemade syringe pump continuously pushes a sample mixture of a specific volume in a glass syringe at a fixed flow rate through a Teflon capillary tube connected to the inlet of the chip. The Arduino Nano controls a continually running motor which applies a continuous force upon the plunger end of the syringe.

### 2.4. Development of the Power Supply and User Interface

A DC power of 12.6 V supported by three serial-connected 18650 batteries is shown in Figure 7a and utilized to supply the power requirement when the system is working. The 18650 cells with a nominal voltage of 3.7 V and a maximum voltage of 4.2 ± 0.05 V are exploited and have a discharge capacity greater than 2.05 ampere-hour (Ah). There is another choice of 12 V external DC power for our device, presented in Figure 7b. The panel DC power jack is an electrical connector to supply the external power for a period of measuring time.

Figure 7c presents the user interface on the device. We develop an interface that sets the operational parameters by pressing the buttons and displays the current status in the LCD 1602A (OpenHacks, Rosario, Santa Fe, Argentina). The steps are as follows. The user can set the denaturation temperature, annealing temperature, and the pumping rate of the sample. When the high- and low-temperature regions reach the steady state, the device does not proceed until the user presses the enter button.

### 2.5. Surface Treatment of the Microchannel

Due to the high surface-to-volume ratio of the PDMS microchannel, a serpentine channel design increases the possibility of enzymes in PCR mixtures sticking onto the hydrophobic surface walls. Any enzyme adsorbed by the walls of the channel reduces its efficiency. Thus, the surface of the PDMS channel is coated with Tween 20 to prevent the adsorption of DNA polymerase. Immediately after the bonding process, the PDMS channel is filled with the 20% Tween 20 solution and kept still for 1 h. This treatment is static surface modification. After removing the 20% Tween 20 solution and rinsing the PDMS channel briefly with 0.25% Tween 20 solution at the flow rate of 1.5 μL/min, the injection of airflow of 3 μL/min cleans out the channel.

Countless bubbles might also form in the denaturation zone and block the microchannel. The generation of air bubbles in the sample solution under high-temperature conditions is one of the crucial topics in microfluidic systems. A volume amount of 50 μL of highly viscous mineral oil with a high boiling point flows into the microchannel just before the introduction of the sample solution, which helps increase the pressure of the sample solution in the microchannel. Then, the latter sample solution follows the flowing mineral oil into the high-temperature zone without air bubbles. A volume of 10 μL of mineral oil injects into the microchannel after the sample solution to prevent the dead volume of the reaction mixture.

### 2.6. Operation of the Device

The DNA sample preparation has been described considerably in our previous work [32,33]. A segment of DNA about 200 bps is amplified. The reaction mixture for CFPCR requires 25 μL of a mixed solution, which contains 5 μL of 20 ng/μL of the DNA template, 1 μL of 0.4 μM of each forward and reverse primer, 10 μL of 5 × PCR-Mastermix (5 × reaction buffer B (400 mM Tris-HCl pH 9.4~9.5 at 25 °C, 100 mM (NH_4_)_2_SO_4_, 0.1% *w*/*v* Tween 20), 12.5 mM MgCl_2_, 1 mM of each dATP, dGTP, dCTP, and dTTP, and *Taq*-DNA-Polymerase) and 9 μL of nuclease-free water. Two primers (5′-TGGAGAGAAATGGGCTACA-3′ and 5′-CATGGTTTTGTGTAATATTGTGA-3′) are used as forward and reverse primers for PCR, respectively.

For our device, we set the temperature zones at 369 K for denaturation and 333 K for annealing after modifying the channel surface, sealing the PDMS surface with a film (Microseal ‘B’ PCR Plate Sealing Film, Bio-Rad, Hercules, CA, USA) and fixing the chip by a clamp. Pre-mixed samples are transported to the microchip using a homemade syringe pump. Then, the reagent moves through sequential temperature zones corresponding to denaturation, annealing, and extension zones.

PCR amplification is also performed with a conventional thermal cycler (MJ Mini™ 48-Well Personal Thermal Cycler, Bio-Rad, Hercules, CA, USA). The cycling conditions are as follows: 40 cycles of 30 s at 367 K for denaturation, 30 s at 333 K for annealing, and 30 s at 345 K for extension; 300 s for pre-denaturation and 420 s for pro-extension are added to the initial and the final steps of PCR, respectively.

The PCR products are collected in a pipette tip from the channel outlet and mixed with 1× blue dye. PCR amplified fragments are analyzed after fractionation by agarose slab gel electrophoresis (Mini-Sub Cell GT System, Bio-Rad, Hercules, CA, USA). Ten μL of each sample are loaded onto 2% agarose gel (Certified Molecular Biology Agarose, Bio-Rad, Hercules, CA, USA) and electrophoresed in 10× Tris/Boric Acid/EDTA (TBE) buffer. After electrophoresis, the gel is stained with 10 mg/mL ethidium bromide solution (Bio-Rad, Hercules, CA, USA) and visualized by UV transillumination.

## 3. Research Methodology

We have fabricated a fully integrated device for continuous-flow PCR. The denaturation zone is in the central part of the chip. The annealing and the extension zones are on opposite sides. Five temperature regions are located within the chip width of 25 mm. First of all, we measure the flow volume of the syringe pump and express the characteristic results of the homemade pump. Then, we investigate the influences of the operational parameters of the system on the temperature uniformity of the chip.

### 3.1. Characteristic Results for the Motor

Our syringe pump features a software interface where a user can control the device to change the flow rate of the sample mixture. The flow rates for the liquid volumes in the glass syringes (25, 50, and 100 μL, 700 series, Hamilton Co., Reno, NV, USA) are measured. Table 1 demonstrates the measured flow rate at various setting values of RPMs with three syringes. We average three experimental results to obtain the measured flow rate at one RPM value. It is convenient for users to set the required flow rate rather than the RPM value. Table 2 presents the setting values of revolution(s) per minute (RPM) at various flow rates with three syringes. The linear relationship between the setting value of RPM and flow rate is acceptable.

### 3.2. Temperature Measurements

Temperature uniformity is one of the critical issues influencing the amplified efficiency of PCR. In the CFPCR device, the isothermal area and the mixture flow rate determine the heating time for the DNA sample. Two main approaches ensure the mixture temperature suitable for CFPCR. Firstly, an infrared (IR) camera (TAS-G100EXD, Nippon Avionics Co., Ltd., Tokyo, Japan) characterizes the spatial temperature distribution across the surface of the PDMS-based chip and evaluates the performance of the thermal modules. After reaching a steady-state temperature distribution, the IR imager captures the IR images of the chip surface with the various sample flow rates.

Figure 8 demonstrates the influence of different sample flow rates on the temperature distributions of the chip surface. The results present temperature gradients between different temperature zones. The five temperature regions become clearly distinct and separate strictly from each other when the water flow rate is 1, 2, 3, 4, and 5 μL/min, respectively, as shown in Figure 8a. Due to the symmetric arrangement of the heating blocks, the symmetry of the temperature distribution on the chip is predictable. When the liquid flows through the serpentine microchannel, a smaller Nusselt number (Nu) of less than one corresponds to increased heat conduction rather than active convection. The water flow rate of less than 5 μL/min results in a small Nu of less than one, and the symmetry of the temperature distribution is distinct. It also shows that the present thermal arrangement is effective in generating the temperature requirement of the CFPCR chip.

The water flow rate is 10, 15, 20, 25, and 30 μL/min, respectively, as shown in Figure 8b(a–e) Due to the high convective effect resulting from the large Nu, the temperature near the denaturation region is lower than the required temperature, and the temperature near the annealing region is higher than the requisite one. The distortions of the symmetry of the temperature distributions are apparent. Inappropriate temperature distribution inside the CF device for PCR may cause either diffuse smearing upon gel electrophoresis or poor DNA amplification efficiency. The temperature distributions in the CFPCR device do not fit with the requisite PCR temperatures.

Secondly, the channel temperatures are measured using thermocouples inserted into the PDMS/glass bonding chip, placed in contact with the glass substrate, and connected to a data acquisition system (Model NI 9211, National Instruments, Austin, TX, USA). Some K-type thermocouples (outer diameter of 0.254 mm, K30-2-506, Watlow Electric Manufacturing Co., St. Louis, MI, USA) measure the mixture temperatures. We immerse the thermocouple in a known temperature inside a water bath to calibrate it before usage. The marked points on the PDMS substrate are the locations of the measuring points. After reaching a steady state, the LabVIEW (National Instruments, Austin, TX, USA) program records the temperatures of the thermocouples at seventeen points shown in Figure 9a.

We can obtain the surface temperatures of the chip with the sample flow rate of 2 μL/min by the infrared imager. The locations of the marked points at the annealing, extension, and nearby regions in Figure 9b are similar to the measuring points in Figure 9a. The measured temperatures lined at the same horizontal positions are averaged and plotted in Figure 9c. From Figure 9c, the inner temperatures refer to the temperatures measured by the thermocouples, and the surface temperatures are the temperatures derived from the IR imager. The setting temperatures of the cartridge heater and Peltier element at the denaturation and annealing regions are 105 °C and 60 °C, respectively. We can find the surface temperatures of 87.3 °C and 58.5 °C and the inner temperatures of 94.8 °C and 55.3 °C near the denaturation and annealing regions. We can use the results to modify the setting temperatures to obtain the proper temperatures for PCR.

## 4. Results and Discussion

The present paper conducts some experimental work to complete the contents of this article. A portable CFPCR device integrated with Arduino boards for thermal controlling and sample pumping amplifies the DNA fragment of *Colla corii asini* successfully.

### 4.1. Preparation Works for CFPCR

We compare the performance of PCR amplification of the microfluidic device with that of the conventional method. A moderate amount of the mineral oil flows into the microchannel before the introduction of the PCR solution. Then the latter solution follows the flowing mineral oil into the high-temperature zone without generating air bubbles. Figure 10 demonstrates the preparation work for using the portable device for CFPCR. By injecting a small amount of black ink into the bonding chip, we can make sure that the channel pattern of the fabrication microfluidic chip is acceptable for CFPCR. The bonding result of the PDMS chip and glass substrate shows excellent leakage-free performance in Figure 10a. We insert two thermocouples onto the annealing zone (Low Temperature Zone) and denaturation zone (High Temperature Zone) of the CFPCR chip. The measured temperatures in Figure 10b show the chip temperatures reaching the steady state within one minute and the variation of temperature at the steady state is less than ±1 °C.

### 4.2. DNA Sample Amplification and Gel Electrophoresis Results

The following figures present the results of gel electrophoresis analysis of the DNA products. We carry out the amplification of a 200 bp DNA fragment on the portable device as well as the commercial PCR machine. The first lane (Lane Mk) indicates the DNA ladder in Figure 11a. The negative control without a DNA template at Lane NC expresses that the amplified product is specific inside a conventional PCR machine. The positive control inside a conventional PCR machine presents at Lane M as reference. The PCR product from the CFPCR chip is at Lane T. Using the image processing software (ImageJ, Version 1.50d, National Institutes of Health, Bethesda, MD, USA) to analyze the images, we obtain the fluorescence intensities in Figure 11b. Despite the slightly weaker PCR band intensity of Lane T than half that of Lane M, they are sufficient for evaluations and analyses in diagnoses. The results in Figure 11 demonstrate that the specific amplification products in our device and the thermal cycler are clear.

The flow rate through the microchannel determines the residence time of a fluid element in a specific temperature zone. Speeding up the flow rate can reduce the total reaction time. It might, however, cause the reaction to be inefficient due to short reaction time in the working zone. PCR performed with various flow rates from 0.3 μL/min to 1 μL/min is shown in Figure 12. The chip system amplifies the DNA product well at a flow rate of 0.3 μL/min. When the flow rate increases, the PCR amplification time is decreased. However, the amount of amplified fragment decreased with higher flow rates from 0.3 μL/min to 0.6 μL/min. This decrease in PCR efficiency may be partly due to the short residence time in the extension zone at a high flow rate, which leads to insufficient time for DNA polymerase synthesis. Consequently, PCR products decreased with the flow rate. Our CFPCR device reduces the run time from 2 h for a conventional PCR down to 69 min at the flow rate of 0.4 μL/min. The device can further reduce its run time down to less than 55 min at the flow rate of 0.5 μL/min.

The CFPCR chip performs the experiments using various initial DNA concentrations for 20 ng/μL, 2.0 ng/μL, 0.2 ng/μL, and 0.02 ng/μL. Figure 13 shows the effects of various initial DNA concentrations (20 ng/μL, Lane D; 2.0 ng/μL, Lane 10^−1^; 0.2 ng/μL, Lane 10^−2^; 0.02 ng/μL, Lane 10^−3^) on PCR amplification when the DNA mixture flows through the microchannel at the flow rate of 0.4 μL/min. The negative control without a DNA template at Lane NC expresses that the amplified product from the CFPCR chip is specific. It demonstrates that the amount of CFPCR products almost decreases with the initial DNA concentration from 20 ng/μL to 1000 × dilution, as shown in Figure 13b. The initial concentration for accomplishing the PCR process is at least 20 ng/μL in the portable CFPCR device. There are some limitations in our CFPCR chip device. Firstly, the outer dimensions of the PDMS layer are 40 mm in length and 25 mm in width. It is not easy to further reduce the volume of the chip. Secondly, the cycle number of our CFPCR reactor is fixed once the design is finalized. Obeid and Christopoulos [34] provided a CFPCR chip and collected the amplification products through the output access holes after cycles 20, 25, 30, 35, and 40. Their chip has the flexibility to change the thermal cycle numbers. However, the seal of the unused output access holes is an issue during PCR.

## 5. Conclusions

People eat food not only for preventing hunger but also for sustaining the body’s well-being. Food security, therefore, is always the first priority when going shopping. *Colla corii asini* is one of the most valuable tonic traditional Chinese medicines. To establish an effective and applicable method to distinguish authentic TCMs is essential. This work describes a portable nucleic acid amplification device integrated with thermal control and liquid pumping connecting to Arduino boards. We present a PDMS/glass bonding chip with a serpentine microchannel of various widths. The cartridge heater supports the thermal energy to sustain the denaturation temperature at the central part of the chip. The Peltier element supplies the energy to withstand the annealing temperature at two sides of the chip. It does reduce the total volume of the chip device due to the symmetrical five-temperature-region pattern, and requires only one Arduino Mega board for the thermal control. We assemble a miniaturized liquid pump and program an Arduino file to push the sample mixture into the chip to implement the PCR process. A DC power supported by three serial-connected 18650 batteries or another 12 V external DC power supplies the power for measuring. We build a portable CFPCR chip device with a compact modular design and the device is suitable for the usage in the remote area. In the proposed operation, the Nusselt number of the sample flow is less than one, and the heat transfer is conduction only. Then, we can ensure temperature uniformity in specific reaction regions. A *Colla corii asini* DNA segment of 200 bp is amplified in the microchannel under the treatment of specific concentrations of Tween 20 solutions to evaluate the PCR performance. To our knowledge, our group is the first to introduce Arduino boards into the heat control and sample pumping modules for a PCR device. The initial concentration for accomplishing the PCR process is at least 20 ng/μL at the flow rate of 0.4 μL/min in the portable CFPCR device. In the future, we can apply the portable continuous-flow device to a low-cost PCR system. Once we finalize the design, low-cost polycarbonate chips may be a good alternative for microfluidics chips. This device not only tests *Colla corii asini* but also can be used for other TCMs. Our future work will optimize the volume of the sample mixture based on the design of experiments. We will also be reporting in future work on an experimental investigation in order to perform a comprehensive analysis of the amplification efficiencies using more than one sample at once in the reaction chip during PCR processes. To further reduce the system cost, we will utilize one Arduino-compatible device with proper programming for controlling the temperature of the heating blocks and a continually running motor. To make the user interface friendlier, the user can set the RPM value by pushing “+” and incrementing 0.1 RPM or pushing “−” and decrementing 0.1 RPM in our future work. Furthermore, we can integrate the battery charger circuit and an ESP32 board into our device to make it more customer-oriented.

## Figures and Tables

**Figure 1 micromachines-13-01289-f001:**
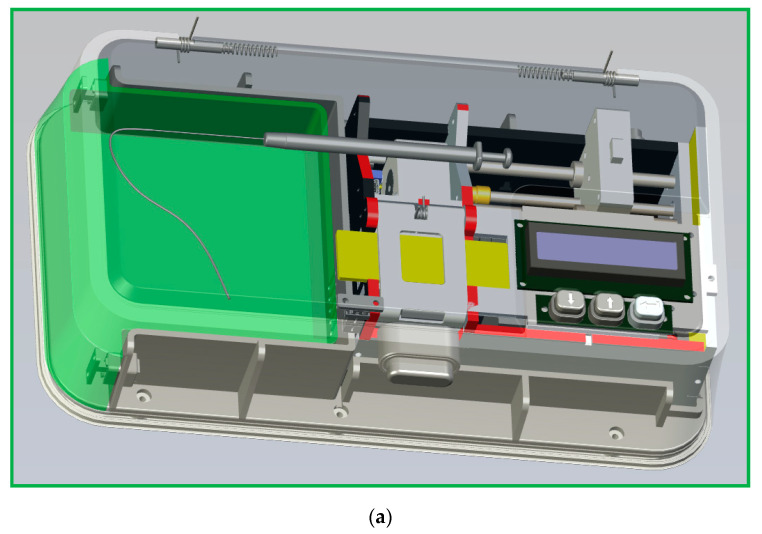
(**a**) A schematic diagram of the portable CFPCR device. Photographic images of the portable CFPCR device (**b**) with outer casing and (**c**) without outer casing.

**Figure 2 micromachines-13-01289-f002:**
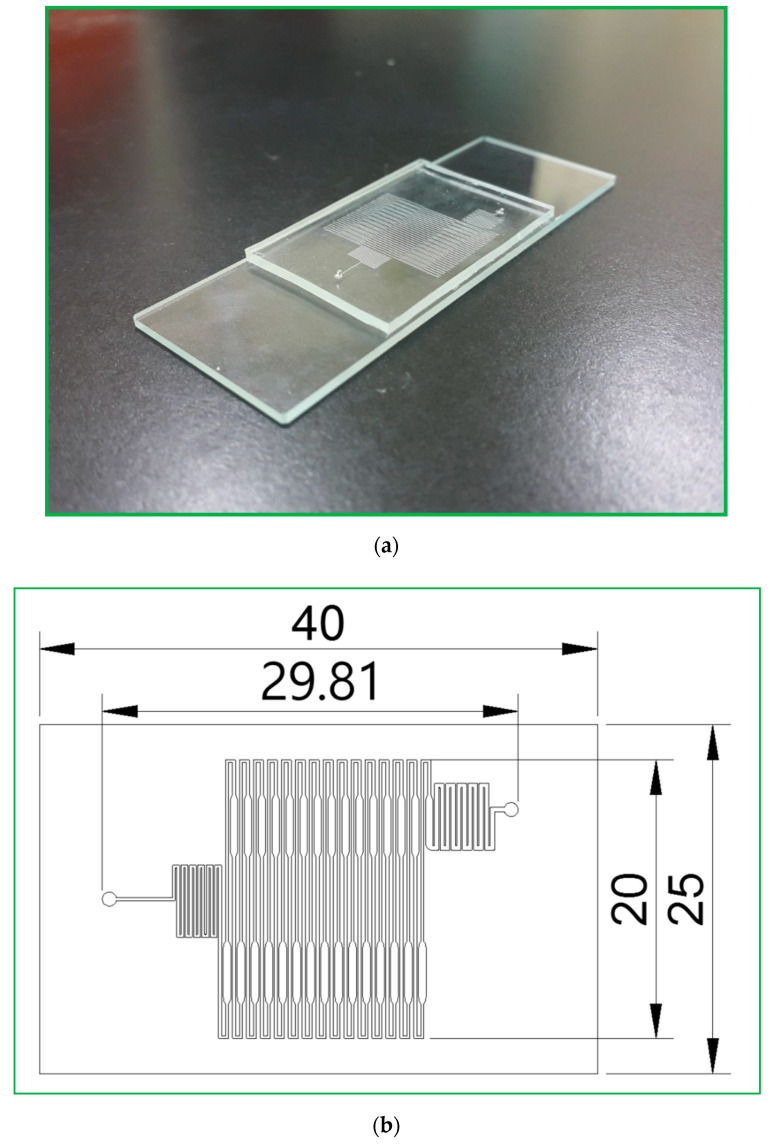
(**a**) A photographic image of the CFPCR chip comprising a PDMS layer and a glass slide. (**b**) A schematic diagram of the 30-loop channel. (**c**) The pre-denaturation part (left) and the pro-extension part (right).

**Figure 3 micromachines-13-01289-f003:**
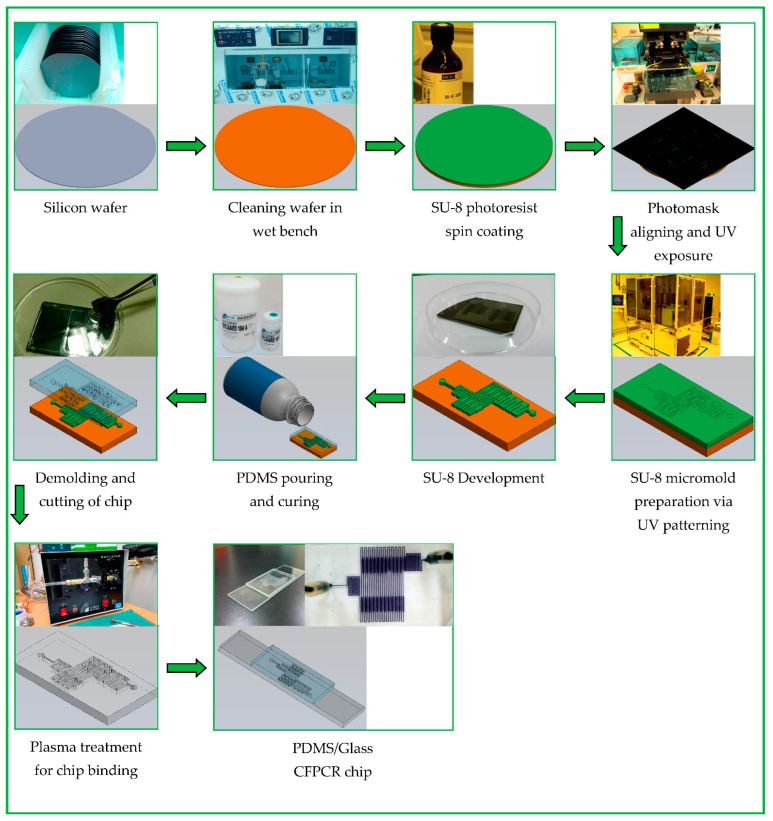
A photographic image and schematic diagram of the fabrication process of the CFPCR chip.

**Figure 4 micromachines-13-01289-f004:**
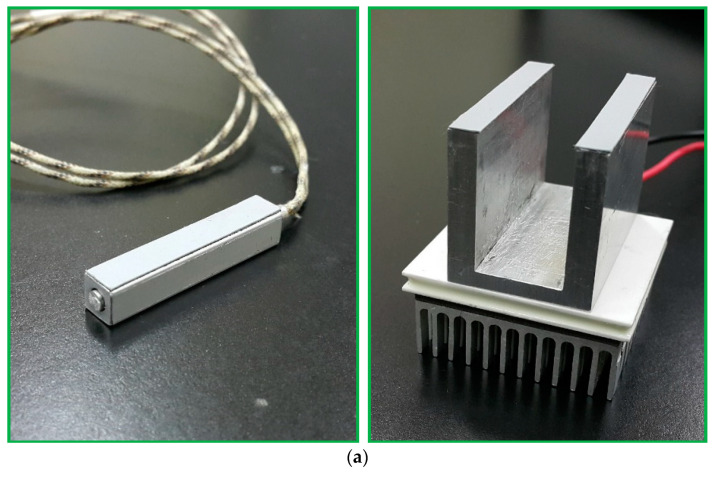
(**a**) The structure of the thermal cycling apparatus consists of a cartridge heater, a Peltier element, aluminum fins, and thermally conductive aluminum blocks. (**b**) The aluminum heating blocks and the PDMS-based chip are assembled and fixed onto a PMMA housing. (**c**) The five-temperature-region design requires only one half-loop per PCR cycle.

**Figure 5 micromachines-13-01289-f005:**
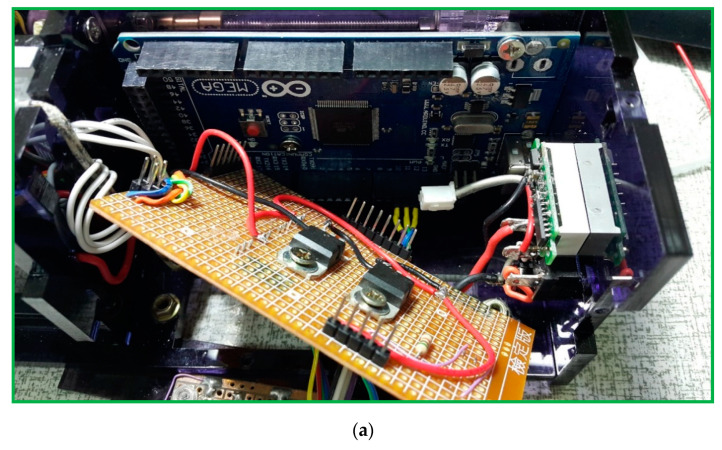
(**a**) A photograph of the thermal control module. (**b**) A schematic diagram of the thermal control functions. (**c**) The five-temperature-region design requires only one half-loop per PCR cycle.

**Figure 6 micromachines-13-01289-f006:**
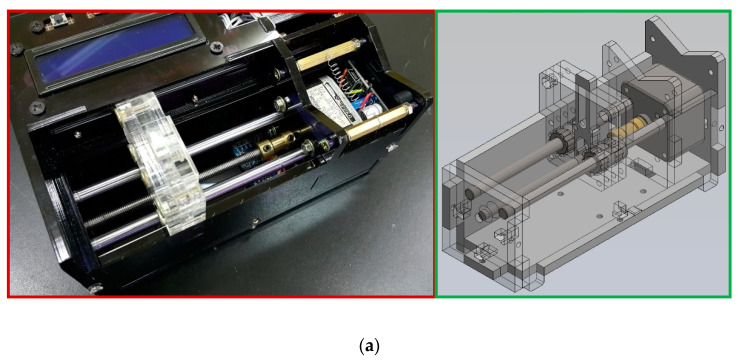
(**a**) A photograph of the homemade pumping unit composed of a stepper motor, a lead screw, a coupling, two linear bearings, and two guide rods. (**b**) A schematic diagram of the block diagram. (**c**) The circuit diagram of the syringe pump based on Arduino.

**Figure 7 micromachines-13-01289-f007:**
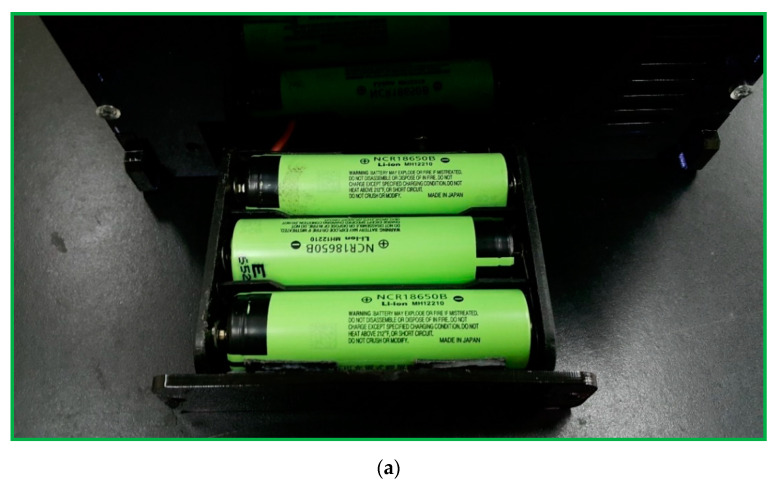
(**a**) A photograph of A DC power of 12.6 V supported by three serial-connected 18650 batteries. (**b**) A choice of 12 V external DC power. (**c**) The user interface sets the operational parameters by pressing the buttons and displays the current status in the LCD 1602A.

**Figure 8 micromachines-13-01289-f008:**
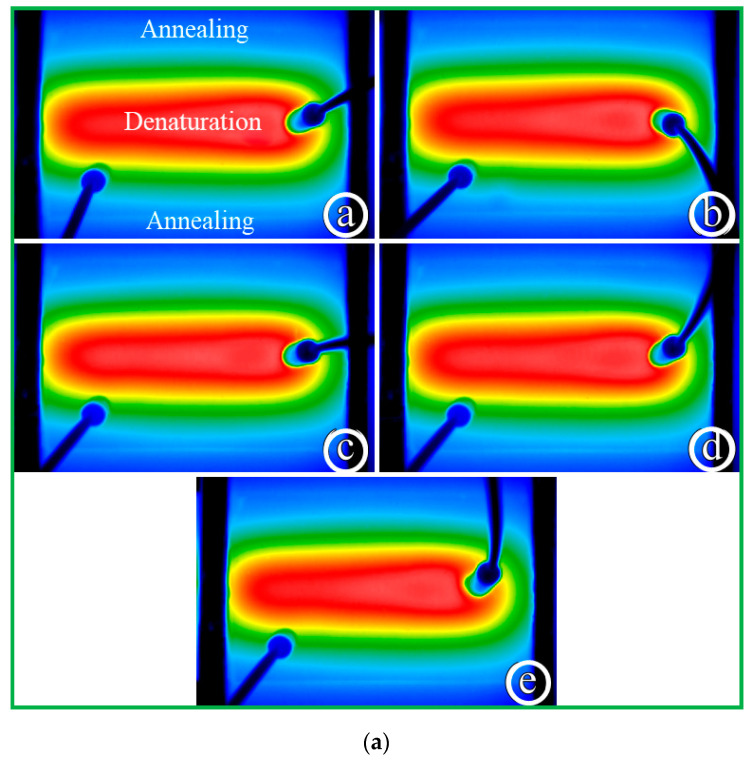
The temperature distributions of the chip surface (**a**) when the water flow rate is 1, 2, 3, 4, and 5 μL/min, respectively, and (**b**) when the water flow rate is 10, 15, 20, 25, and 30 μL/min, respectively.

**Figure 9 micromachines-13-01289-f009:**
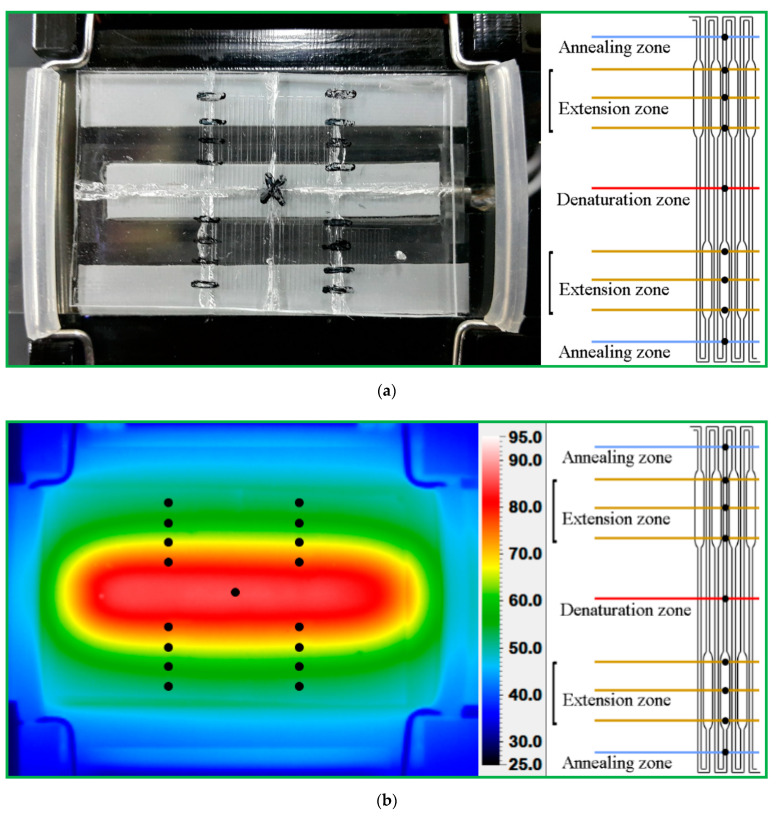
(**a**) The measured locations of the thermocouples at seventeen points. (**b**) The locations of the marked points at the annealing, extension, and nearby regions by the IR imager. (**c**) The measured temperatures lined at the same horizontal positions.

**Figure 10 micromachines-13-01289-f010:**
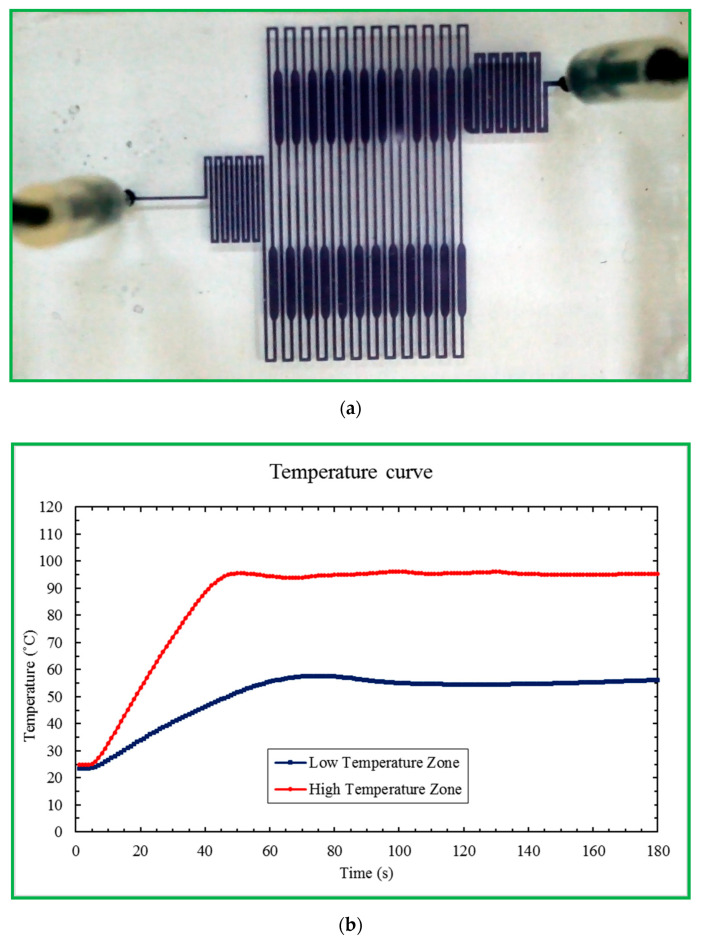
(**a**) The channel pattern of the fabrication microfluidic chip. (**b**) The measured temperatures at the annealing zone (Low Temperature Zone) and denaturation zone (High Temperature Zone) of the CFPCR chip.

**Figure 11 micromachines-13-01289-f011:**
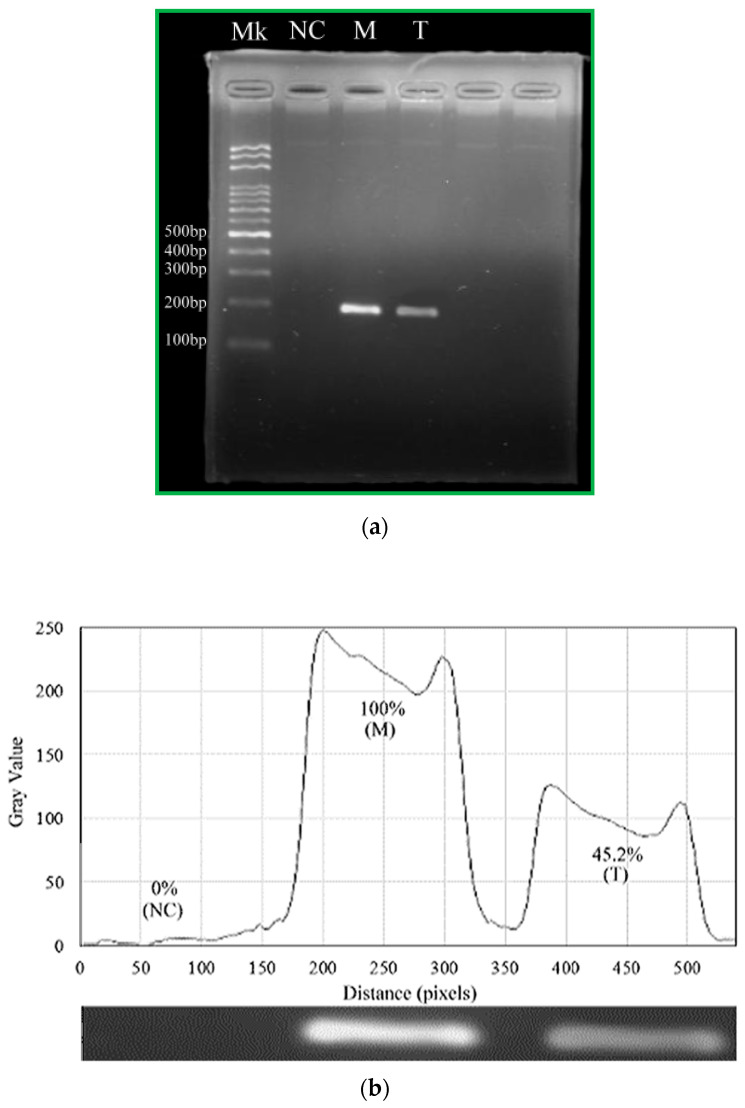
(**a**) The results of gel electrophoresis analysis of the products. The first lane (Lane Mk) indicates the DNA ladder. The second lane (Lane NC) indicates the negative control inside a conventional PCR machine. The 200 bp PCR product in the commercial PCR machine and CFPCR chip (Lanes M and T). (**b**) The grey intensities of PCR products by image analysis.

**Figure 12 micromachines-13-01289-f012:**
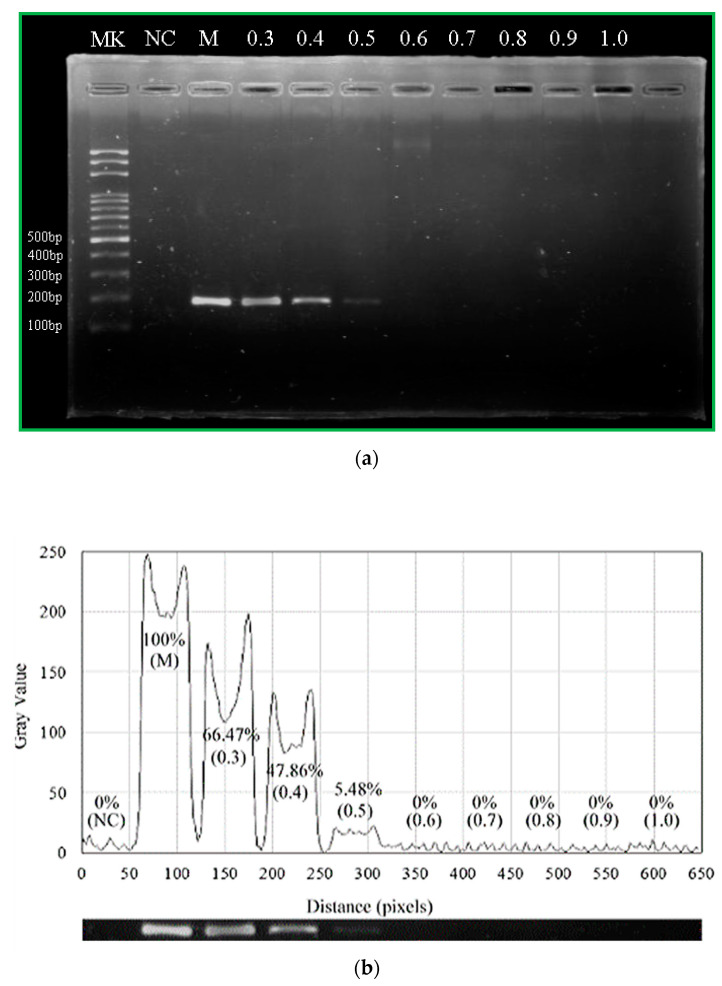
(**a**) The results of gel electrophoresis analysis of the products. The first lane (Lane Mk) indicates the DNA ladder. The second lane (Lane NC) indicates the negative control. The 200 bp PCR product in the commercial PCR machine and CFPCR chip (Lanes M and the others). (**b**) The grey intensities of PCR products by image analysis.

**Figure 13 micromachines-13-01289-f013:**
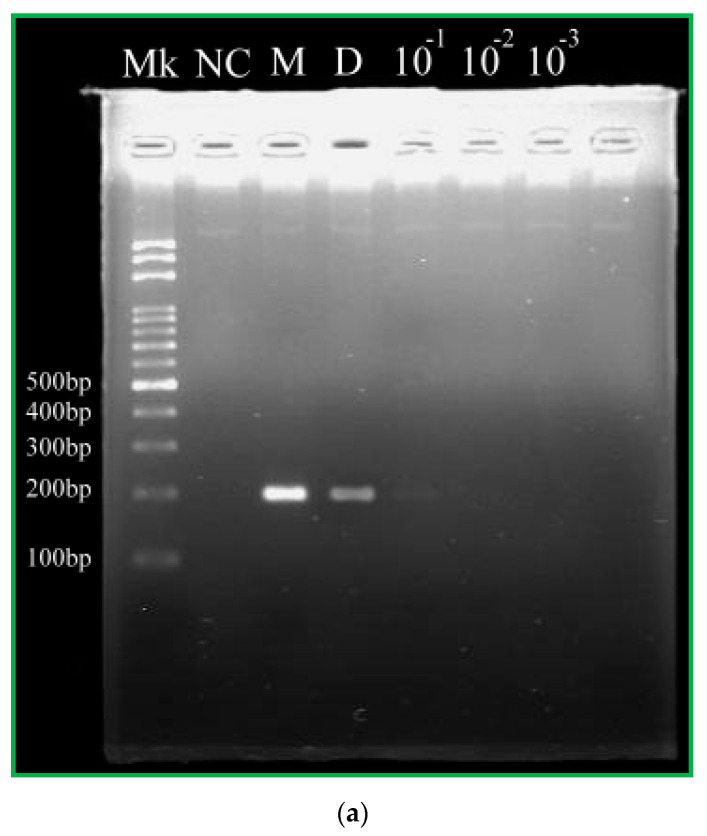
(**a**) The results of gel electrophoresis analysis of the products. The first lane (Lane Mk) indicates the DNA ladder. The second lane (Lane NC) indicates the negative control from the CFPCR chip. The 200 bp PCR product in the commercial PCR machine and CFPCR chip (Lanes M, D, and the others). (**b**) The grey intensities of PCR products by image analysis.

**Table 1 micromachines-13-01289-t001:** The measured flow rates at various setting values of RPMs with three syringes.

	Flow Rate (μL/min)		
RPM ^1^	25 μL	50 μL	100 μL
1	0.034	0.066	0.134
2	0.068	0.132	0.268
3	0.102	0.198	0.402
4	0.136	0.264	0.536
5	0.17	0.33	0.67
6	0.204	0.396	0.804
7	0.238	0.462	0.938
8	0.272	0.528	1.072
9	0.306	0.594	1.206
10	0.34	0.66	1.34

^1^ The results of the setting RPM up to 75 RPM are not shown here.

**Table 2 micromachines-13-01289-t002:** The setting values of RPMs at various flow rates with three syringes.

	RPM		
Flow Rate (μL/min) ^1^	25 μL	50 μL	100 μL
0.1	3	NA	NA
0.2	6	3	NA
0.3	9	NA	NA
0.4	12	6	3
0.5	15	NA	NA
0.6	18	9	NA
0.7	21	NA	NA
0.8	24	12	6
0.9	27	NA	NA
1.0	30	15	NA

^1^ The results of the setting flow rate up to 10 μL/min are not shown here.

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
