# Peer review of "A Portable Continuous-Flow Polymerase Chain Reaction Chip Device Integrated with Arduino Boards for Detecting Colla corii asini"

_micromachines, 2022, doi:10.3390/mi13081289_

Round 1

Reviewer 1 Report

The authors have introduced a portable CF-PCR chip device for the detection of fake Colla corii asini. This is a very important step in traditional Chinese medicine because it is an efficient way to detect authentic Colla corii asini. However, I have a few questions.

1. Does this device only test Colla corii asini or it can be used for other TCMs?

2. The conventional PCR can run more than one sample at once, is this the case with the CF-PCR chip device or it can only run one sample at once?

3. Is the run time for your CF-PCR chip device comparable to that of a conventional PCR or does it require less time to complete?

4. Line 216 states that the reactor chip is disposable. Is the reactor chip what holds the sample in during the PCR process which is FIgure 2a? Please state this clearly. 

5. Are there any limitations in your CF-PCR chip device? Please include that in the last paragraph of section 3

Reviewer 2 Report

Below are the comments and suggestions which, when appropriately addressed by the authors, may enhance the quality of the paper:

1. The Introduction section is not clear. The research question or motivation of the authors should be stated clearly. As the last paragraph of the section, please discuss the structure the different parts of the manuscript.

2. Please discuss the Figures and tables in an elaborate fashion. Please check the spelling typo mistakes found in the figures (e.g. SMART instead of SMAR). Please arrange scalability of tables.

3. Add a section called Research Methodology and present the details extensively. Please discuss the methodology used in this study. Please specify if this study is a survey. If yes then please describe the survey, the criteria of selecting the related studies, which period and which search engine and keywords.

Reviewer 3 Report

This manuscript described a full integrated by Arduino boards portable continuous-flow PCR chip. Although the development, miniaturization and control of the chip components have been reported previously, its integration and control by using an Arduino board is quite original and very useful for the monitoring of the ongoing covid pandemic. The chip's fabrication, assembly and control are detailed in extensive detail, including significant useful information for the development of devices based on this technology.

Therefore, I consider this outstanding article deserves publication in this Journal after addressing the following minor concerns:

1.       Detail the different components of the CFPCR device in Figure 1 a and b

2.       The chip photo show in figure 3 with the blue colored solution don’t match with the actual chip design

3.       The amplification platform should be tested for non-complementary DNA templates to ensure the specificity of the methodology.

4.       Are the chips for single use? if it is possible to reuse them, please detail how the system is cleaned between samples.

5.       Discuss the chip fabrication reproducibility

6.       Detail the volume of sample used for amplification, the volume of amplification product obtained and the dead volume of the system.

This is a paper with enough novelty, quality, and relevance to deserve its publication in this Journal after addressing the minor issues pointed out.

Reviewer 4 Report

The authors introduce an Arduino-based method to control a portable continuous-flow polymerase device. The implementation is demonstrated by detection of Colla corii asini DNA. The article is interesting and generally well-written however I have some questions and remarks.

A) Within the abstract the author claim: “To our knowledge, our group is the first to introduce Arduino boards into the heat control and sample pumping modules for a CFPCR device”. I see a slight contradiction with the introduction, where ref [26] and [27] are already using Arduino-compatible devices.

B) The authors use two Arduino-board (Arduino MEGA and Arduino Nano) to control the system. I do not see the reason for using more than one board, with proper programming a single board would be more than enough.

C) Page 10 line 292: “A single intelligent temperature sensor (DS18B20, Dallas semiconductor, Dallas, Texas, USA) provides the temperature value for feedback control.”

If I am correct, two sensors were used in the setup. How was the output of these sensors used?

D) Figure 5. The figure shows 4 AAA batteries while the device is powered by 3 LiIon batteries. The cabling of the 5 V supply is a bit weird. The push buttons do not have pull-up resistors. Are internal pull-ups used? The Peltier-element and the heater get the unregulated supply voltage. Since this voltage changes over time, it can cause problems.

E) Page 12 line 338: “The ULN2003A adjusts the motor speed according to the command given by the Arduino Nano”. The ULN2003A is a simple transistor array, it does not perform any speed adjustment. The Arduino Nano is in charge of controlling the speed an outputting the correct sequence for driving the coils within the stepper motor.

F) Figure 6 (b): There is a typo: Stepper Moter instead of Stepper Motor

G) Figure 6 (c): The connection of ULN2003A is not correct, the stepper motor would not work in this configuration. Note: the ULN2003A driver can be used only with unipolar stepper motors (with at least 5 wires). The common pin of the driver should also be connected (to the supply voltage). Note: there is one extra battery showed in the picture.

H) Table 2. It is not clear why some flow rates cannot be configured with different syringes.

I) A charger and a cell balancing circuit would be useful in the device.

J) In my opinion logging the process parameters would be quite useful if the device would be used to determine the quality of any product. The device should have a communication interface for transferring the collected data.

Round 2

Reviewer 4 Report

The authors answered most of the issues. I have the following remarks:

Figure 6 (c): The connection of ULN2003A is still not totally correct, the outputs of the Arduino are connected to Input 1 to 4 (pin 1 to 4 on ULN2003A), the output 1 to 4 should be used (pin 16 to 13 should be used).

Chapter 4. Conclusion: Instead of “power recharger circuit” I would call it “battery charger circuit”.
